# Enteral Nutrition in Operated-On Gastric Cancer Patients: An Update

**DOI:** 10.3390/nu16111639

**Published:** 2024-05-27

**Authors:** John K. Triantafillidis, John Papakontantinou, Pantelis Antonakis, Manousos M. Konstadoulakis, Apostolos E. Papalois

**Affiliations:** 1Department of IBD and Endoscopy, “Metropolitan General” Hospital, 15562 Holargos, Greece; jktrian@gmail.com; 2Hellenic Society of Gastrointestinal Oncology, 15562 Athens, Greece; 32nd Department of Surgery, School of Medicine, Aretaieion Hospital, University of Athens, 10676 Athens, Greece; johnpapacon@hotmail.com (J.P.); dr.antonakis@gmail.com (P.A.); mkonstad@med.uoa.gr (M.M.K.)

**Keywords:** gastric cancer, enteral nutrition, nutrition, preoperative nutrition, postoperative nutrition

## Abstract

It is well established that the preoperative nutritional status of gastric cancer (GC) patients significantly affects the prognosis of the operated patients, their overall survival, as well as the disease-specific survival. Existing data support that preoperative assessment of nutritional status and early correction of nutritional deficiencies exert a favorable effect on early postoperative outcomes. A variety of relevant indices are used to assess the nutritional status of GC patients who are candidates for surgery. The guidelines of almost all international organizations recommend the use of oral enteral nutrition (EN). Oncologically acceptable types of gastrectomy and methods of patient rehabilitation should take into account the expected postoperative nutritional status. The majority of data support that perioperative EN reduces complications and hospital stay, but not mortality. Oral EN in the postoperative period, albeit in small amounts, helps to reduce the weight loss that is a consequence of gastrectomy. Iron deficiency with or without anemia and low serum levels of vitamin B12 are common metabolic sequelae after gastrectomy and should be restored. EN also significantly helps patients undergoing neoadjuvant or adjuvant antineoplastic therapy. The occurrence of the so-called “postgastrectomy syndromes” requires dietary modifications and drug support. This review attempts to highlight the benefits of EN in GC patients undergoing gastrectomy and to emphasize the type of necessary nutritional management, based on current literature data.

## 1. Introduction

Despite the apparent international decline in the incidence of gastric cancer (GC), it still remains a major health problem with morbidity and mortality ranking in the top five, worldwide. A characteristic clinical feature of this neoplasm is the gradual onset of symptoms, which results in delayed diagnosis and, consequently, a worsening of the prognosis. Weight loss and malnutrition are among the most alarming clinical signs. It is well known that the preoperative nutritional status significantly affects the prognosis of the operated patients with GC, their overall survival, and the disease-specific survival. Today, malnutrition is considered to be one of the most important factors predisposing to postoperative complications and poor prognosis of GC patients undergoing gastrectomy [1,2]. However, we should bear in mind that in many Western cohorts, 40–50% of patients with CG do not undergo surgery due to advanced disease, while in Eastern countries, 40% of patients may have early GC that does not have a similar risk of malnutrition.

The prevalence of malnutrition in GC has been estimated to be around 60% preoperatively, although it varies greatly depending on the stage of the tumor, the type of treatment applied, and the methods of nutritional assessment used in the individual patient [3]. Using different tools of nutritional assessment, Ryu et al. found that the prevalence of malnutrition at admission of GC patients was 31% using SGA and 43% using the NRS-2002 tool. Six months postoperatively, a good correlation between the nutritional assessment tools and the other nutritional measurement tools was found. However, 12 months after surgery, most patients who were assessed as malnourished using SGA and NRS-2002 had returned to their preoperative status [4]. Wang et al. reported that 52.5% of 81 cases with GC undergoing subtotal gastrectomy and 20 cases undergoing total gastrectomy had, or were at risk for, malnutrition with symptom severity, employment status, and difficulty in diet preparation being the most significant predictors of nutritional status, indicating a need for continued monitoring and support after discharge from hospitals [5]. In the study by Lee et al., 21.4% of patients still suffer from malnutrition 1 year after surgery [6]. Guo et al., using the PG-SGA tool, found that among 2322 hospitalized Chinese patients with GC, 80.4% were malnourished and 45.1% required urgent nutritional support. Old age, female, residence in a village, a lower level of education, and self-paying were risk factors of severe malnutrition [7]. Therefore, malnutrition of hospitalized patients with GC in China is very common.

The impact of malnutrition relates not only to the deterioration of quality of life and overall survival but also to the occurrence of adverse postoperative complications and conditions such as infections, hospital stays, and side effects of chemotherapy. Other symptoms such as anorexia, dysphagia, abdominal pain, nausea, and vomiting, as well as dumping syndrome, exocrine pancreatic insufficiency, mucositis, and diarrhea seem to be the main etiopathogenic factors of malnutrition. It is well accepted that preoperative assessment of nutritional status and early correction of nutritional deficiencies exert a favorable effect on early postoperative outcomes.

In this review, the authors attempt to highlight the benefits of perioperative EN in GC patients undergoing gastrectomy and to emphasize the type of necessary nutritional management, based on current literature data.

## 2. Gastrectomy for Gastric Cancer: Clinical Consequences and Treatment

The consequences of gastrectomy on the overall physiology of the patient are quite important if one considers that the removal of the stomach automatically implies the removal of the “store” of food consumed, there is no mechanical digestion, and there are no mechanisms for the promotion of food to the intestine. The consequences of these changes are related to the occurrence of the various “postgastrectomy syndromes”. Therefore, there has recently been a tendency to remove as little of the stomach as possible; however, this is performed without harming the intended therapeutic effect, which is the complete removal of the tumor itself. For example, in cases of early GC, laparoscopic gastrectomy with pylorus preservation is regularly performed; an operation that achieves satisfactory results in terms of patient survival and adequate postoperative nutrition [8]. Laparoscopic gastrectomy is superior to open gastrectomy not only because of the less pain, less use of analgesics, early mobilization, early restoration of intestinal peristalsis, and shorter hospital stay, but also because of the better effect on postoperative nutritional status [9]. However, differences between the two interventions do not seem to be important in terms of postoperative short-term nutritional status and terms of weight loss [10].

The main clinical consequences of surgical removal of the stomach performed due to GC can be summarized as follows.

### 2.1. Weight Loss

Weight loss is an important symptom of cancer patients with a variety of effects. It has been estimated that patients with gastroesophageal cancer lose approximately 10% of their body weight in the 6 months before cancer diagnosis. Continuous weight loss is observed in operated-on GC patients during the first postoperative year, which stabilizes thereafter. Weight loss has been correlated with the extent of gastric resection. The causes of weight loss are multiple and include anorexia, diarrhea, restriction of food intake, and malabsorption.

Weight loss before surgery has long been evaluated in daily clinical practice. Body mass index (BMI) is an index of fitness, but its relationship with surgical outcome is not always favorable to the patient. For example, Chen et al. found that low BMI is associated with more severe postoperative complications and worse prognosis [11]. In the same study, it was also noticed that despite a higher risk of mild postoperative complications, patients with a high BMI had paradoxically better survival compared to that of patients with a normal BMI [11]. In another study, Yang et al. found that a visceral fat area was an independent risk factor for postoperative complications. They have shown that the determination of a visceral fat area represents a superior predictor compared to BMI in terms of the short-term effects of obesity [12]. It seems that unintentional weight loss is a factor of poor prognosis and mortality in patients with GC. In the study of Cui et al., among 672 cases with GC, no weight loss was observed in 275 cases, limited weight loss was observed in 294 cases, and severe weight loss was observed in 103 cases. There were significant differences between the three groups as far as tumor size, location, depth of invasion, the number of lymph node metastasis, surgical approach, extent of lymphadenectomy, and curability was concerned. The 5-year survival rate of the patients with severe weight loss, limited weight loss, and no weight loss was 28.0%, 37.7%, and 40.3%, respectively. In multivariate analysis, age, weight loss before surgery, depth of invasion, and node stage were independent prognostic factors for survival. Patients with weight loss above 10% have poor prognosis. They concluded that weight loss before surgery may be an independent prognostic factor for patients with GC [13]. Fukahori et al. evaluated the incidence of weight loss in 131 patients with advanced GC after the initiation of chemotherapy, as an indicator of the presence of cachexia and the relationship of weight loss to overall patients’ survival. Weight loss was defined as loss of weight greater than 5% or loss of weight greater than 2% in those patients with a body mass index less than 20 kg/m^2^ within the last 6 months after chemotherapy initiation. The median age and median Eastern Cooperative Oncology Group performance status of the patients were 68 years and 0, respectively. The incidence of weight loss was 53% in the first 12 weeks, increasing to 88% after 48 weeks. The overall survival rates were significantly correlated with weight loss at 12, 24, and 48 weeks. Interestingly, weight loss was observed within 12 weeks of starting chemotherapy in 50% of patients [14]. Weight loss was significantly associated with adverse events and reduced survival, which highlights the importance of monitoring weight loss as well as providing nutritional support at the start of chemotherapy.

### 2.2. Cachexia

Cachexia is an irreversible condition characterized by a significant loss of body weight, muscle mass, and adipose tissue accompanied by the presence of many metabolic, hormonal, and immunological factors. However, even today, the underlying pathophysiological mechanisms are not sufficiently known. It is more prevalent in male GC patients than in females, probably due to the difference in the muscle composition between the two sexes. Its incidence is higher in more advanced stages of cancer. It seems certain that the key role of the stomach in digestive processes and appetite regulation is an additional aggravating factor for its occurrence. Cachexia significantly affects the recovery of GC patients [15]. Understanding the mechanisms related to the occurrence of this condition at preclinical and clinical stages could further improve the available treatment options.

### 2.3. Sarcopenia

Sarcopenia is a term referring to a loss of skeletal muscle mass. It occurs in a significant proportion of patients with advanced GC strongly affecting chemotherapy tolerance, surgical complications, tumor recurrence, and survival [16]. Lidoriki et al. showed that patients with low skeletal muscle mass index were older, had lower serum albumin levels, and had a lower BMI, suggesting that skeletal muscle mass is substantially related to the nutritional status of GC patients [17]. It appears that preoperative skeletal muscle mass is a useful nutritional predictor tool of postoperative complications and survival of GC patients [18]. The findings of a systematic review and meta-analysis showed higher morbidity, lower survival, and higher in-hospital mortality in GC patients with preoperative sarcopenia [19], a finding that has also been associated with an increase in infectious complications after gastrectomy. Park et al. found that GC patients undergoing gastrectomy had a continuous and progressive decrease in skeletal muscle mass. They also showed that fat mass decreased in the first year but subsequently recovered [20]. Preoperative exercise and nutritional support programs have been shown to improve sarcopenia and postoperative outcomes in patients with advanced age GC and sarcopenia [21].

### 2.4. Maldigestion and Malabsorption

The nutritional elements that are malabsorbed are numerous and their clinical consequences are quite significant. The nutrients that are malabsorbed primarily include vitamin B_12_ and iron as a consequence of gastric acid deficiency and duodenal bypass. The type of surgical procedures performed on GC patients varies depending on many factors such as the location of the tumor, the general condition of the patient, etc. These types of operations include a Billroth I and Billroth II surgical resection, a Roux-en-Y gastrojejunostomy, and a total gastrectomy with Roux-en-Y esophagojejunostomy. However, the Billroth I procedure is now rarely performed in most countries [22].

Malabsorption can manifest itself with a wide range of symptoms ranging from complete absence to abdominal distension, flatulence, diarrhea, and severe steatorrhea. The main incriminating mechanisms include pancreatic exocrine failure (up to 75% of cases) and bacterial overgrowth [23]. The mechanisms of pancreatic insufficiency include a decrease in pancreatic secretion due to the absence of gastric reflexes and asynchrony between the arrival of nutrients in the gut and the pancreatic secretory response [24]. The values of fecal elastase underestimate the presence of pancreatic exocrine fatty acid deficiency in gastrectomized patients but the triolein-labeled mixed triglyceride breath test (13C) seems to be a rather useful test [25]. Empirical treatment with pancreatic enzymes at a dose of 50,000 IU lipase at main meals, always depending on the severity of symptoms and the fat content of the diet, is claimed to help. However, the available studies on pancreatic enzyme supplementation in GC patients describe conflicting results [26]. For example, Sridhar et al. suggested that routine pancreatic supplementation after gastrectomy may not be necessary. However, they added that if the symptoms of exocrine pancreatic insufficiency become worse, one needs to perform appropriate investigations followed by pancreatic enzyme replacement therapy [27].

The occurrence of small intestinal bacterial overgrowth after gastrectomy is common with a prevalence fluctuating between 63% and 78% [28]. The mechanisms are related to the loss of hydrochloric acid and its bactericidal activity, impaired gastrointestinal motility, and a decrease in defense molecules in intestinal secretions [28]. There are few studies dealing with the predominant clinical symptoms in patients with SIBO, the majority of them suggesting that the most common symptom caused by SIBO is diarrhea, followed by abdominal pain and bloating [28]. Diagnosis is made by culture of intestinal fluid taken from the small intestine or by appropriate breath tests. Antibiotics such as rifaximin, metronidazole, and ciprofloxacin are also given for a certain period [29]. If the dyspeptic symptoms are adequately managed, the patient is not placed on a diet poor in fat. Other dietary restrictions include avoiding a very high fiber intake, because fiber may reduce pancreatic enzyme function, increase fat malabsorption, and trigger symptoms. In cases in which bacterial overgrowth of the small intestine and pancreatic exocrine insufficiency do not respond to conservative treatment, dietary fat is replaced with medium chain triglycerides oil (MCT oil).

### 2.5. Vitamin B_12_ Deficiency

Due to the lack of intrinsic factor due to gastrectomy, there is a malabsorption of vitamin B_12_, which is estimated to be 100% in patients who underwent total gastrectomy and about 16% in patients who underwent partial gastrectomy 4 years after surgery [30]. Prophylactic administration of B_12_ should be started in the days following total gastrectomy, while its administration after partial gastrectomy should be based on serum levels, taking into account that elderly individuals have additional small bowel bacterial overgrowth despite the absence of predisposing factors. Moreover, patients with low preoperative vitamin B_12_ levels are at greater risk of developing this deficiency. The recommended dose for patients who have undergone gastrectomy is 1000 μg intramuscularly per month, although its administration is effective even after oral intake [31]. Kim et al. showed that oral vitamin B_12_ replacement therapy in patients undergoing total gastrectomy for GC is safe and effective [32]. The interpretation of the efficacy of the oral administration is based on the existence of a second transport system for vitamin B_12_ that does not require the presence of the endogenous agent or intact terminal ileum but is absorbed via passive diffusion.

### 2.6. Calcium and Vitamin D Deficiency

It has been estimated that among patients who have undergone gastrectomy, 45% have low levels of vitamin D 25-OH and 76% develop secondary hyperparathyroidism [33]. Plasma calcium levels are usually normal due to calcium mobilization from the bones. Some authors recommend prophylactic calcium and vitamin D supplementation in GC patients who have undergone gastrectomy [33]. Other micronutrients such as folic acid, zinc, and fat-soluble vitamins should be administered if deficiencies are found during follow-up.

### 2.7. Iron Deficiency and Anemia

Iron deficiency is common in patients with GC after surgery. However, iron deficiency and anemia are also common at the time of diagnosis, therefore being among the most important alarming laboratory findings. The causes are related to duodenal bypass, digestive losses, and malabsorption due to hypochlorhydria. At least 40% of patients require iron supplementation. The prevalence of anemia after gastrectomy approaches 24% and is due to iron and vitamin B_12_ deficiency, and possibly folic acid. Blood losses from the anastomotic site or due to bacterial overgrowth in blind loops are also possible causes of anemia. However, reduced iron absorption due to hypochlorhydria and duodenal bypass represents the major contributor to iron deficiency, because reduced gastric acidity impedes the conversion of Fe^3+^ to the Fe^2+^ form, which is more readily absorbed [34]. In a retrospective study of 119 patients who underwent distal gastrectomy with Billroth I or Roux-en-Y reconstructions for stage I GC, it was shown that the Roux-en-Y reconstruction was the unique responsible risk factor for hemoglobin reduction [35]. Treatment of iron deficiency anemia aims to restore the hemoglobin value to normal, replenish iron stores, and address the potential cause of blood loss. The administration of iron sulfate or gluconate is usually effective. It is administered at a dose of 150–300 mg/day in divided doses. Intravenous administration is recommended in cases of intolerance to oral formulations.

### 2.8. Bone Disease

The incidence of osteoporosis after gastrectomy occurs at 36%, is higher in women, and is independent of the extent of gastric resection. It is mainly attributed to secondary hyperparathyroidism due to inadequate intake and malabsorption of calcium and vitamin D. Patients are advised to have an adequate intake of calcium and vitamin D and periodic monitoring of bone density [36].

### 2.9. Postgastrectomy Syndromes (Dumping Syndrome)

Dumping syndrome refers to gastrointestinal and vasomotor symptoms, the appearance of which is a consequence of the rapid propulsion of the hyperosmotic content toward the small intestine [37]. It can be divided into early or late depending on the clinical symptoms. Early dumping syndrome occurs within the first hour after eating and is manifested by gastrointestinal (fullness, abdominal pain, nausea, diarrhea) and vasomotor (sweating, palpitations, flushing) symptoms. Late dumping syndrome, which is less common, occurs 1–3 h after food intake and mainly causes vasomotor symptoms due to hypoglycemia, which is a consequence of hyperinsulinemia due to the rapid arrival of food in the intestine. Because of the symptoms, patients avoid eating, thus promoting weight loss and reducing their quality of life. Mine et al. studied the prevalence of dumping syndrome in 1153 patients operated on for GC [38]. They found that 67.6% of patients experienced at least one symptom of early dumping, most commonly abdominal pain or fullness, while 38.4% experienced at least one symptom of late dumping. It is largely expected that patients who underwent total gastrectomy had the most symptoms.

The treatment of dumping syndrome is based on patient information and strict adherence to dietary advice such as the adoption of small and frequent meals, good chewing of food, excess protein intake, avoidance of processed carbohydrates, avoidance of concurrent intake of liquid and solid foods, and avoidance of alcohol consumption. Pharmacological therapy is reserved for patients in whom conservative means fail. Acarbose (an alpha-glucosidase inhibitor) at a dose of 50–100 mg before meals is effective because it reduces carbohydrate absorption and therefore the incidence of hypoglycemia in late dumping syndrome. Somatostatin analogs can also be administered subcutaneously (three times daily) or intramuscularly every 2 or 4 weeks as extended-release formulations. 

## 3. Evaluation of the Nutritional Status of Gastric Cancer Patients

The assessment of the nutritional status of patients with malignant disease is of particular clinical importance. The assessment of the current nutritional status, the ability to feed, and the severity of the underlying disease should be performed regularly starting at first contact and continuing at short intervals (e.g., every 4–8 weeks) to identify as early as possible any decrease in the level of nutritional status. This assessment is particularly useful for hospitalized elderly patients because poor nutritional status is associated with an increased rate of postoperative complications and increased length of hospital stay. A number of screening tools have been developed for the early detection and management of malnourished GC patients. A brief description of the utility of the available methods is given below.

### 3.1. Biochemical Factors

Various hematological and biochemical factors such as albumin, rapidly metabolized proteins (proalbumin, retinal-binding protein), C-reactive protein, total cholesterol, cholinesterase, glucose, hemoglobin, and neutrophil and lymphocyte count are some of the indices whose estimation of preoperative level is necessary. Various markers and scoring systems have been developed to identify patients with poor nutritional status. These systems have been successfully used to predict the occurrence of postoperative complications and to estimate possible survival [39].

### 3.2. Anthropometric Parameters

The assessment of muscle mass and fat should be performed with the help of special instruments. The estimation of the Karnofsky index is also recommended. Assessment of nutrient intake and changes in BMI should be carried out at the time of diagnosis of cancer and then at regular intervals. Assessment of muscle mass and fat stores may be performed by dual X-ray absorptiometry or bioimpedance analysis. In a retrospective study of 775 patients undergoing gastrectomy for GC, it was found that patients with a BMI less than 18.5 and low preoperative albumin levels had significantly reduced overall survival after gastrectomy [40].

### 3.3. Skinfold Thickness (SFT)

The SFT is estimated quite accurately utilizing a special instrument—the skinfold caliper (Figure 1). A disadvantage of the instrument is the fact that measurements are impossible to be made by the patient himself, but only through the help of another person.

The thickness of the skin folds is measured at four points on the body, namely the back and front of the upper arm and the back under the shoulder blade and the side of the waist (Figure 2). We then calculate the sum of the four measurements and compare the results with the values in the tables available in order to estimate the amount of body fat.

### 3.4. Mid Arm Circumference

This parameter is regularly used to estimate the loss of muscle mass. If it is less than 20 cm or if it decreases by 2 cm between two determinations, it indicates malnutrition.

### 3.5. Screening Tools and Questionnaires

#### 3.5.1. Nutritional Risk Screening 2002 (NRS)

The NRS tool is a quite simple and useful tool for assessing nutritional risk in hospitalized patients, especially the elderly. Using this tool, patients are divided into those who are at increased nutritional risk and those who are not. In particular, increased scores on the NRS tool are associated with increased rates of postoperative complications and increased length of hospital stay [41].

#### 3.5.2. Malnutrition Screening Tool (MST)

The MST is composed of two questions: weight loss and food intake/appetite. Based on the results, patients could be classified as patients “at risk for malnutrition” or patients “without risk of malnutrition”. A weight loss of 10% for 6 months, or 5% during 3 months, is considered the most reliable indicator of nutritional deficit. This tool has more predictive value in elderly patients. Chen et al. investigated the possibility of detecting the presence of cachexia in 1001 patients who underwent elective radical gastrectomy for GC using the four most commonly used nutritional screening tools: the Malnutrition Universal Screening Tool (MUST), the Nutritional Risk Screening (NRS)-2002, the Malnutrition Screening Tool (MST), and the Short Nutritional Assessment Questionnaire (SNAQ). The results were compared with the international consensus diagnostic criteria for cancer cachexia. They found that the MST had the greatest ability to detect cancer cachexia among patients with GC [42].

#### 3.5.3. Patient-Generated Subjective Global Assessment (PG-SGA)

The PG-SGQ tool assesses both clinical information and physical examination findings. The tool takes into account the existence of weight loss, the clinical history, and certain analytical data. It requires the cooperation of the patient, who fills in questions about their type of symptoms, diet, and daily physical activity. Patients are finally classified as patients with normal nutritional status, patients with moderate malnutrition, and patients with severe malnutrition. According to Cho et al., the timing of the nutritional evaluation may be important in identifying and treating malnutrition related to GC prognosis [43].

### 3.6. Prognostic Nutritional Index

The prognostic nutritional index (PNI) has been widely used because of its effectiveness, simplicity, and convenience and is calculated by the following formula: 10 × serum albumin value (g/dL) + 0.005 × peripheral blood lymphocyte count. A value of less than 45 is indicative of severe nutritional impairment, whereas a value greater than or equal to 45 is associated with a normal nutritional status. In a related meta-analysis of GC patients, PNI was associated with depth of invasion and lymph node metastasis [44]. Sakurai et al. showed that in patients with GC, preoperative PNI is an independent predictor of both overall survival and disease-specific survival. In particular, patients with stage 1 and 2 UICC disease had significantly worse outcomes in the low PNI group than in the high PNI group [45].

### 3.7. Short Nutritional Assessment Questionnaire (SNAQ)

The SNAQ is an easy, short, valid and reproducible questionnaire for early detection of hospital malnutrition. In a relevant study Harada et al. found a significant difference in event-free survival when the SNAQ scores were classified into two groups (i.e., scores of ≤3 (SNAQ3) or scores of ≥4 (SNAQ4)). They noticed that SNAQ4 was associated with a higher risk of undernutrition. Therefore, SNAQ is a predictor of undernutrition in cancer patients receiving outpatient chemotherapy. It is suggested that patients with an SNAQ score of equal or greater of 4 receive dietary guidance at an early stage [46].

### 3.8. Other Tools

Based on the available 25 nutrition-screening tools, the Chinese Society of Nutritional Oncology built a nutrition-screening tool named age, intake, weight, and walking (AIWW). According to the described results, the AIWW tool showed a better nutrition-screening effect than NRS-2002 and MST for cancer patients. The authors suggest that this nutrition-screen tool could be recommended as an alternative nutrition-screening tool for the cancer population [47].

In summary, several tools for the estimation of nutritional risk and malnutrition are available, with the PNI possibly representing the most important one in everyday clinical practice. The PNI, or the combination of a preoperative BMI less than 18.5 and low albumin levels, is predictive of decreased overall survival after gastrectomy. In a recent systematic review and network meta-analysis of 16 studies including 5695 patients, the authors evaluated the diagnostic accuracy of the available preoperative nutritional screening tools for adult patients undergoing surgery and identified the test with the highest accuracy. The nutrition screening tools were as follows: Malnutrition Screening Tool (MST), Malnutrition Universal Screening Tool (MUST), Mini Nutritional Assessment (MNA), short-form Mini Nutritional Assessment (MNA-SF), Nutritional Risk Index (NRI), Nutrition Risk Screening 2002 (NRS-2002), and Preoperative Nutrition Screening (PONS). They found that MUST had the highest overall test accuracy performance (sensitivity 86%, specificity 89%). The Network meta-analysis showed that the NRI had similar sensitivity but lower specificity than the MUST. According to the authors, the predictive accuracy of the MUST index does not justify the implementation of interventions aimed at optimizing nutrition, without taking into account other tests [48].

## 4. Enteral Nutrition (EN)

The term EN includes oral supplementation feeding (peroral and via a tube; e.g., percutaneous endoscopic gastrostomy, radiologic gastrostomy, and endoscopic jejunostomy). The energy requirements of GC patients are considered similar to those of healthy individuals (25–30 kcal/kg/day). Regarding protein requirements, they range between 1 and 1.2–1.5 or 2 g/kg/day; however, in patients with chronic renal failure, the amount of protein should not exceed 1 g/kg/day. The lipid/carbohydrate ratio shall be determined according to the clinical condition of the individual patient. In the presence of peritoneal carcinomatosis or if there is obstruction or ascites, water and sodium intake should be less than normal (30 mL/kg/day for water and 1 mmol/kg/day for Na). In patients undergoing major surgery, preoperative malnutrition is associated with increased morbidity and mortality [49].

Nutritional support in patients with GC is indicated when malnutrition is present, the patient is unable to obtain food, and food intake is less than 60% of daily requirements. It is recommended for all patients with inadequate dietary intake for more than 2 weeks. It is also recommended for both GC patients undergoing surgery and those with unresectable disease. The intervention aims to improve nutritional status, metabolism, and compliance with administered antineoplastic therapy, quality of life, and disease progression. After gastrectomy, a diet based on frequent small meals with a restriction of simple carbohydrates is recommended to avoid dumping syndrome.

EN maintains the structural and functional integrity of the gastrointestinal tract. It represents the most suitable dietary choice for patients with dysphagia or obstruction when oral intake does not meet nutritional requirements. It is safer, cheaper, and more physiological than parenteral nutrition. However, in patients with impaired gastrointestinal function, parenteral nutrition is mandatory. Although parenteral nutrition provides optimal nutrition, it increases the risk of infections compared to EN. Therefore, in the late stages of the disease, the benefit of nutritional support is rather small and associated with an increased risk of complications. In general, nutritional support should be applied when the benefits outweigh the potential complications or when patients wish to have it.

### 4.1. Perioperative (Pre and Post) Enteral Nutrition of Patients Operated on for Gastric Cancer

For patients undergoing surgery for GC, preoperative malnutrition is associated with increased morbidity (i.e., increased infection rate, delayed wound healing, and pulmonary complications, including adult respiratory distress syndrome) and mortality. Thus, improving the nutritional status before surgery could improve the postsurgical outcome of GC patients. In an intact gastrointestinal tract, EN is as effective as parenteral nutrition. Therefore, an appropriate assessment of the preoperative nutritional status and subsequent nutritional intervention before gastrectomy is essential for malnourished patients with GC. EN may be performed through a nasogastric or a nasoenteric tube for short periods, whereas direct access to the bowel such as via a jejunostomy should be preferred if EN is given for more than 3 weeks [50]. Independently of nutritional status, the Enhanced Recovery after Surgery (ERAS) group has provided recommendations for the perioperative holistic management of patients with upper gastrointestinal tumors. They recommend a preoperative carbohydrate loading (800 mL of a 12.5% carbohydrate drink the night before surgery and 400 mL the following morning 2 h before induction of anesthesia) to reduce the insulin resistance and tissue glycosylation caused by the surgery, help in postoperative glucose control, and sustain normal bowel function [51].

Following the previously mentioned premises, which indicate prioritizing oral and enteral optimization over parenteral optimization, it is recommended that nutritional support be started without delay in malnourished patients and in those whose requirements are not being met. The duration of nutritional support must be at least 7–14 days before the procedure. It has been shown that oral nutritional support in severely malnourished patients with GC during the perioperative period decreased the incidence, severity, and duration of postoperative complications. Moreover, preoperative carbohydrate loading during the night before surgery reduces the insulin resistance and tissue glycosylation caused by the surgery, helps in postoperative glucose control, and sustains normal bowel function. 

The importance of postoperative nutritional support in GC patients lies in maintaining as normal a nutritional status as possible in the postoperative catabolic period. A prospective study of 435 GC patients found that the prevalence of severe malnutrition increased significantly after surgery (2.3 and 26.3% before and after surgery, respectively) [52]. It appears that older age, preoperative weight loss, and open surgery are the most important risk factors for severe postoperative malnutrition. Postoperatively, food intake due to reduced appetite is also reduced. The subsequent abnormal nutritional status may take up to a year to begin to recover. However, small bowel functions recover between 6 and 12 h after surgery, which supports that EN may begin during this time. Gabor et al. showed that starting EN 6 h after surgery is quite safe [53].

The importance of early enteral nutrition (EEN) in the clinical course of GC patients has been demonstrated in several studies. (Table 1). Li et al. separated 400 GC patients undergoing total gastrectomy into two groups of 200 patients each. Patients in the control group received postoperative parenteral nutrition (PN), while patients in the experimental group received postoperative ΕEN. Clinical outcomes, immunological parameters, and nutritional status of the patients were evaluated. Patients who received EΕN had significantly shorter episodes of pyrexia, time of intestinal function recovery, exhaust time, and length of hospital stay compared to the control group. The activities of CD3^+^, CD4^+^, CD4^+^/CD8^+^, and NK cells were significantly lower in both groups on the first postoperative day compared with preoperative levels. After treatment, the levels of CD3^+^, CD4^+^, CD4^+^/CD8^+^, and NK cells in the EEN patients were similar to preoperative levels, whereas in the control group patients, the immune cell levels were significantly lower compared with preoperative values. These findings suggest that the widespread use of EEN should be universally accepted [54]. In addition to the significant level of safety that it offers, it was found that early oral EN via a nasogastric tube after surgery for GC does not increase the incidence of postoperative complications compared to EN [55]. These favorable effects of early EN had already been noted in the 2000s. In a later study, Hur et al. found that early EN after surgery for GC leads to shorter hospital stays and improves many parameters of patients’ quality of life postoperatively [56]. Similar results were obtained by Barlow et al. [57]. In another randomized clinical trial of 105 patients undergoing surgery for digestive cancer, early postoperative EN reduced high metabolism, preserved intestinal barrier function, and reduced the incidence of intestinal infections [58].

Perioperative EN also reduces the incidence of postoperative complications in malnourished GC patients. In a related study, 468 patients with GC or colorectal cancer with moderate or severe malnutrition were randomized into two groups: Group A with seven-day preoperative EN followed by 7-day EN and Group B without dietary intervention (control group). The authors observed a twofold reduction in postoperative complications and a threefold reduction in deaths in patients with perioperative EN with a predominantly impressive reduction in minor septic complications and a reduction in hospitalization days [59].

Another study showed that preoperative application of EN in GC patients improves postoperative nutritional status and immune function while reducing the inflammatory response. This study, which included 200 GC patients, compared the postoperative nutritional status and the inflammatory response after the administration of EN, which started either 1 week preoperatively (study group) or after surgery (control group) for 9 days postoperatively. No significant differences were found in the time of the first rectal gas passage, abdominal distension, blood glucose, liver and renal function, and electrolytes between the two groups. However, the study group showed a significantly better immune response than the control group in terms of the various immunological parameters studied. It therefore appears that preoperative initiation of EN benefits GC patients undergoing gastrectomy [60]. Similar to the previous results, a study of 106 GC patients also showed favorable outcomes, because patients who received preoperative nutritional support showed improved postoperative nutritional status and immunological parameters, facts that reduce the inflammatory response and facilitate patient recovery [61]. A study from Iran also confirmed the favorable effect of early oral feeding after surgery for GC. The authors showed that the operation becomes not only safe without significant adverse effects, but also allows the rapid restoration of normal digestive tract function and the reduction in hospital stay [62]. Early EN may even effectively reduce insulin resistance levels in gastrectomized GC patients [63].

Okamoto et al. suggest that early arginine-enriched EN in GC patients undergoing total gastrectomy may improve the patients’ nitrogen balance. Their study included 19 patients who were divided into two groups as follows: the arginine-rich EN group (10.1 g/day) and the group receiving parenteral nutrition. The two groups received identical amounts of amino acids (54 g/day). The authors found no significant differences between the two groups in terms of postoperative complications, length of hospital stay, nutritional status, and body weight. Nitrogen balance remained negative until postoperative day 7 in the parenteral nutrition group but became neutral on postoperative day 7 in the arginine-rich enteral nutrition group [64]. Therefore, despite the absence of a positive effect on weight loss, it appears that early arginine-rich EN improves nitrogen balance after total gastrectomy in GC patients.

It appears that early EN in GC patients undergoing subtotal or total gastrectomy improves prognosis, although the high intolerance rate seems to be the most important factor preventing postoperative EN administration. He et al. classified 66 GC patients who underwent subtotal or total gastrectomy for GC into two groups as follows: the group which received preoperative oral nutritional supplements (31 patients) and the group that only nutritional advice was given (35 patients). Patients in both groups were fed through a nasogastric tube from the first to the fifth postoperative day. The intolerance rate in the first group was numerically lower than that of the second group but was not statistically significant. The main symptoms of gastrointestinal intolerance were distension of the abdomen and abdominal pain, which did not differ significantly between the two groups, nor did the rates of postoperative nausea, vomiting, reflux, and retrograde burning. The study suggested that short-term preoperative EN did not improve food intolerance in the early stage after gastrectomy for GC [65].

Meng et al. investigated the effect of oral nutritional supplements and dietary advice in 337 patients undergoing gastrectomy for GC. The patients were divided into two groups as follows: 171 patients who received both oral nutritional supplements and dietary advice and 166 who received dietary advice only. The mean daily intake of oral nutritional supplements in the intervention group was 370 mL. Three months after surgery, patients who received oral nutritional supplements and nutritional advice had less weight loss, a higher BMI, and a higher skeletal muscle index compared to the nutritional advice group. Furthermore, the incidence of sarcopenia and changes in the type of chemotherapy administered were significantly lower in the oral nutritional supplements group than in the control group. Finally, some quality-of-life parameters including anorexia and easy fatigue were significantly improved in the group receiving oral nutritional supplements and nutritional advice compared to the nutritional advice group [66]. These findings support the postoperative administration of oral nutritional supplements combined with nutritional counseling in GC patients undergoing gastrectomy.

It is generally accepted that weight loss after gastrectomy for GC is associated with a decrease in quality of life and a poor prognosis. Oral nutritional supplements have long been used to reduce weight loss, which occurs predominantly in the first three postoperative months. Miyazaki et al. classified 880 GC patients undergoing curative gastrectomy into two groups as follows: the group with oral EN with 400 mL of 400 kcal caloric daily for 12 weeks (437 patients) and the control group (443 patients). They found that weight loss during the first 3 months was significantly lower in the EN group than in the control group. However, the difference gradually decreased after 6 months and became nonsignificant 1 year after surgery. The improvement in weight loss persisted for 1 year after surgery in patients receiving more than 200 kcal/day of EN [67].

In a prospective study, Martos-Benítez et al. showed that a gastrointestinal rehabilitation program that included pain relief, early mobilization, antibiotic administration, drug prophylaxis against deep vein thrombosis, and respiratory physiotherapy, and an early postoperative EN program that included gastroprotection, management of postoperative nausea and vomiting, early removal of the nasogastric tube, and early EN in patients undergoing gastrointestinal cancer surgery reduced major complications, respiratory complications, delirium, infections, and gastrointestinal complications. There was also a reduction in mortality, length of stay in the intensive care unit, and length of hospitalization [68].

Diarrhea is a common complication of EN. In a prospective randomized study, Zhao et al. studied the effect of fiber and probiotics in reducing the incidence of EN-related diarrhea in GC patients undergoing gastrectomy. Patients were divided into the following three groups for a total of 7 days after surgery: fiber-free diet (FF group n = 40), fiber-rich diet (FE group n = 40), and fiber-enriched and probiotic-enriched diet (FEP group n = 40) It was found that the number of diarrheal stools was higher in the FF group than in the FE group. The number of diarrheal stools in the FE group was less compared to the FE group. The first flatus time was shorter in the FE group compared to the FF group. Gastrointestinal disorders did not differ in the FE and FF groups, but the FEP group had less gastrointestinal disorders than the FF group. Length of hospital stay in the FE and FEP groups was shorter than that of the FF group. The results of the study suggest that the combination of fiber and probiotics reduces the number of diarrheal stools associated with EN of gastrointestinal patients for GC [69].

Positive results with the use of per os dietary supplement were obtained from a study from Japan. The authors administered Racol^®^ NF (Otsuka Pharmaceutical Factory, Tokyo, Japan), a liquid enteral nutritional formulation, to 82 gastrectomized patients with stage I-III GC as an adjunct to regular meals immediately postoperatively and for 3 months. It was found that the mean rate of weight loss at 3 months postoperatively was 8.3%. A significant correlation was also found between adherence to treatment with an amount of nutritional formulation greater than 200 µL per day and the rate of weight loss [70]. Another study also from Japan investigated the effect of an elemental diet formulation (Elental; EA Pharma Co., Ltd., Tokyo, Japan) on postoperative long-term weight loss. The control group was given a normal diet while the elemental diet group received 300 kcal plus the normal diet for 6–8 weeks. It was found that weight loss 1 year after surgery was significantly lower in the elemental diet group compared to the control group in patients undergoing total gastrectomy but not in those undergoing partial gastrectomy. In the multivariate analysis, the elemental diet was the only factor influencing postoperative weight loss for 1 year after surgery [71].

Little data, contrary to the above, has also been reported. Shimizu et al. found no significant differences in the length of postoperative hospital stay between the intervention and control groups of patients undergoing distal gastrectomy. In the patients undergoing distal gastrectomy, the incidence of postoperative complications was significantly higher in the intervention group compared to the control group. In contrast, the length of postoperative stay was significantly shorter in the intervention group undergoing total gastrectomy. In this study, it was shown that early oral feeding did not reduce the postoperative hospital stay after distal gastrectomy and the higher rate of postoperative complications did not advocate early oral feeding for patients undergoing distal gastrectomy [72]. Interestingly, many surgeons are reluctant to start early EN after total gastrectomy. Sierzega et al. highlighted the safety and feasibility of early postoperative EN even after total gastrectomy [73]. Placement of a jejunal tube should be considered in patients of advanced age who have undergone total gastrectomy, patients with severe preoperative malnutrition, patients who are likely to lose weight postoperatively, and patients at high risk for postoperative complications. Early initiation of EN ensures satisfactory postoperative nutritional management.

In conclusion, EN can be safely initiated 6 h after surgery through a percutaneous jejunostomy tube. Early postoperative nutrition reduces the high metabolism associated with surgical trauma, preserves intestinal barrier function, and reduces the incidence of intestinal infections, thus contributing to patients’ faster recovery.

**Table 1 nutrients-16-01639-t001:** Enteral nutrition clinical studies in patients operated on for gastric cancer.

Author/ Year/Citation	Number of Pts & Controls	Groups	Clinical Outcome/ Hospital Stay	Conclusion
Barlow et al., 2011 [57]	121 (38GC, 29PC, 54EC)	EN vs. controls (nil by mouth)	16 days vs. 19 days	Potential benefit of early oral nutrition.
Hur H et al., 2011 [56]	58	Early EN vs. Late EN	Significant differences were noticed	Early oral feeding: shorter hospitalization. Improved QoL, (early postoperative period).
Yao K et al., 2013 [63]	Total number of pts: 77 Group A: 42 Group B: 35	TPN vsEEN by tubes (250–500 mL 5% NaCl and glucose IV for 24 h followed by EN emulsion from 48 h, and then total EN)	Insulin resistance was present early (days 1 to 7) in GC pts. Significant differences between pts who were operated on and those who had not were found. Insulin sensitivity: higher in group B vs. group A.	EEN alleviates insulin resistance in operated-on GC pts.
Mahmood-zadeh H, et al., 2015 [62]	109 pts	Group A (EOF): (1st postoperative day). Group B (LOF): (Nil by mouth until the return of bowel sounds)	Better clinical outcomes in the EOF group. More common rehospitalization in the LOF group. Gas passage, nasogastric tube discharge, time to start a soft diet, and hospital discharge: earlier in the EOF group.	EOF is safe. EOF is associated with favorable early in-hospital outcomes. EOF is associated with a shorter hospital stay.
Ding D, et al., 2015 [61]	106 pts of GC.	Trial group: preoperative one week EN. Control group: early postoperative EN group	PA and IgG levels of the trial group were higher vs. the control group on the postoperative 10th day. IL-6 level of the trial group: lower vs. control group.	EN support improves the postoperative nutritional status and immune function, alleviates inflammatory response, and facilitates recovery.
Wang F, et al., 2015 [60]	200 pts of GC.	Study group: EN starting 1 week before surgery. Control group: EN starting early after surgery.	No differences in: Time of passage of gas, abdominal distension, blood glucose, hepatic and renal function. Albumin and prealbumin levels decreased 1 day after the operation. IgG: higher in the study group Inflammation indices: lower in the study group.	Preoperative EN support improves postoperative nutritional status and immune function and reduces inflammatory response.
Li B. et al., 2015 [54]	400 pts. 200 in the experimental and 200 in the control group.	Control group: Postoperative parenteral nutrition (PN). Experimental group: Postoperative EN.	Postoperative fever time, intestinal function recovery time, anal exhaust time, and the length of hospital stay for patients in the experimental group: Shorter vs. control group. Activities of multiple immune cell types: Lower in both groups when compared with preoperative levels.	After EEN of pts undergoing radical resection for GC, the clinical outcome, immune function, and nutritional status were significantly improved
Kobayashi D et al., 2017 [70]	82 eligible pts operated on for GC	Racol^®^ NF at a dose of 400 kcal/400 mL/d was started within 7 days postoperatively. Continued for 3 months.	Adherence to Racol^®^ NF therapy was the only factor that correlated with the body weight loss ratio among all clinical characteristics	Racol^®^ NF supplementation: significant reduction in body weight loss for pts who tolerated >200 mL/d.
Zhao R et al., 2017 [69]	120 pts	Group A: Fiber-free (FF) group, n = 40), Group B: Fiber-enriched (FE) group, n = 40, Group C: Fiber & probiotic-enriched (FPE) group, n = 40. Postoperative EN: 7 days in all pts	Diarrhea cases: Higher in FF vs. FE group & lower in FEP vs. FE group. First flatus time: Shorter in FE vs. FF group No differences between the FE and FEP groups. Intestinal disorder cases: Lower in FEP vs. FF group LOHS: Shorter in the FE and FEP vs. FF group.	The combination of fiber and probiotics was significantly effective in Treating diarrhea that is associated with EN in postoperative pts with GC.
Martos-Benítez FD et al., 2018 [68]	Prospective study. 465 pts submitted to GI surgery for cancer and admitted to an oncological ICU	General rules: Pain relief, early mobilization, antibiotic and deep vein thrombosis prophylaxis, respiratory physiotherapy. GI rules: Gastric protection, control of postoperative nausea, early removal of the nasogastric tube, EN	Reduction in major complications, respiratory and infectious complications, GI complications, delirium, ICU mortality, length of ICU stay, length of hospitalization.	The program of GI rehabilitation and early postoperative EN is associated with reduced postoperative complications and improved clinical outcomes in pts undergoing GI surgery for cancer.
Shimizu N et al., 2018 [72]	Pts who underwent DG or TG for GC	Intervention group (EOF) vs. Control group (conventional postoperative management) for DG or TG.	No significant differences in LHS between EOF and the control group (pts with DG). Incidence of postoperative complications: greater in the DG EOF group. In contrast, the LHS was shorter in the TG EOF group.	EOF did not shorten the postoperative hospital stay after DG. The higher incidence of postoperative complications precluded the unselected adoption of EOF for DG pts.
Kimura Y et al., 2019 [71]	106 pts	Control group: Regular diet alone postgastrectomy. ED group: 300 kcal ED plus regular diet for 6–8 weeks on postoperative BWL.	BWL 1 year postoperatively: Lower in the ED group than in the control group among pts undergoing TG, but not in patients who underwent DG. Multivariate analysis: ED is the only factor affecting BWL (pts who underwent TG).	Daily nutritional intervention for 6–8 weeks reduced BWL postoperatively and at 1 year in pts who underwent TG.
Miyalazi Y et al., 2021 [67]	1003 GC pts. BWL data were available in 880 pts (ONS 437, control 443)	Pts were assigned to the ONS or the control group. In the former, 400 mL (400 kcal)/d for 12 weeks as EN was planned, and the actual intake amount was recorded daily by the pts.	After 3 months: BWL: lower in the ONS group vs. control group After 6 months: Difference gradually declined. After 1 year: BWL not significant ONS group: 50.4% of pts took >200 mL/d of ONS and showed less BWL at 1 year than the control	ONS for 12 weeks after gastrectomy did not improve BWL at 1 year. The improvement in BWL remained until 1 year after surgery in pts who took more than 200 kcal/d of ONS.
Meng Q et al., 2021 [66]	353 pts at nutritional risk (171 in the ONS group and 166 in the control group)	ONS with dietary advice or dietary advice alone (control) for 3 months after discharge.	After 3 months: ONS and dietary advice group: Less weight loss and higher BMI and SMI vs. dietary advice alone. Sarcopenia: Lower in the ONS group vs. control group. Postoperative chemotherapy in ONS and dietary advice group: Fewer chemotherapy modifications.	The findings strongly support the concept of the introduction of postdischarge ONS with dietary advice to this patient cohort.
He FJ et al., 2022 [65]	66 pts completed the trial (31 in the ONS group, and 35 in the DA group.	Preoperative ONS group vs. DA group. Both groups were fed via NJs (1st day to the 5th day after surgery).	FI rate in the ONS group: Lower than that in the DA group. The postoperative 5-day 50% energy compliance rate in the ONS group was higher than that in the DA group.	Short-term preoperative ONS cannot improve FI and the energy compliance rate in the early stage after radical gastrectomy.
Okamoto Y et al., 2023 [64]	19 pts	Pts were assigned to: PN group and EAN group (for 7 days after surgery).	No differences in: Postoperative complications, LHS, oral intake, BW. Serum arginine levels: similar. Nitrogen balance: Negative up to postoperative day 7 in the PN group and neutral in the EAN group.	Arginine-rich EEN could improve the nitrogen balance after total gastrectomy.

EN = enteral nutrition EEN = early enteral nutrition, EOF = early oral feeding, ONS = oral nutritional supplements, PN = parenteral nutrition, BWL = body weight loss, Pts = patients, FI = feeding intolerance, NJs = nasojejunal tubes, DA = dietary advice, EAN = enteral arginine-rich nutrition, GC = gastric cancer, PC = pancreatic cancer, EC = esophageal cancer, QoL = quality of life, GI = gastrointestinal, ICU = intensive care unit, LHS = length of hospital stay, DG = distal gastrectomy, TG = total gastrectomy, ED = elemental diet.

### 4.2. Nutrition in GC Patients Undergoing Chemotherapy

The side effects of chemotherapy (diarrhea, vomiting, anorexia, mucositis, dysphagia, etc.) often occur in GC patients, resulting in an impairment of the patient’s nutritional status due to the reduction in food intake [74]. Their early detection in patients undergoing chemotherapy at any stage of their disease and treatment is of paramount importance for the patients. Alongside symptomatic treatment, there should be appropriate dietary adjustments according to the type of symptoms and especially to the presence or absence of dysphagia. In cases where the oral route of nutritional support fails, nutritional support via catheter or stoma is indicated. Jejunostomy is indicated in cases of obstructive GC or in individuals who require continuous artificial feeding. Jejunostomy during laparoscopy performed as part of preoperative staging of GC is feasible and effective, although not without complications [75].

Regarding the type of nutritional formulation to be used, the ESPEN guidelines recommend the selection of omega-3 fatty acid–enriched formulations in patients with advanced cancer undergoing chemotherapy with concomitant weight loss, malnutrition, or in patients who are at risk but in whom appetite, lean body mass, and body weight are improved. Parenteral nutrition may be used in cases where EN is inadequate or not feasible, as it is not effective and may be harmful in patients with a functional gastrointestinal tract [76]. 

## 5. Conclusions and Future Perspectives

Early nutritional feeding and effective nutritional intervention through appropriate nutritional screening tools are necessary actions to improve clinical outcomes and reduce the rate of complications. Adequate nutritional management has a potentially positive impact on the clinical status, quality of life, and even survival of patients with GC. Early estimation of nutritional status prevents the occurrence of malnutrition and increases survival rates. Therefore, regular nutritional screening both at diagnosis and throughout the disease, to identify any risk of malnutrition and, if positive, to carry out a full nutritional assessment, is of paramount importance. Outpatients should also be assisted by the primary care physician to identify any existing nutritional problems early and refer them to a specialist nutritionist if required. 

A simple nutritional assessment by taking a brief nutritional history, determination of anthropometric parameters, and basic analytical determination should be able to be adequately performed by the oncologist. Furthermore, the oncologist should have sufficient nutritional training to be able to refer a patient who is at nutritional risk or already malnourished to a nutritionist promptly. There should be close collaboration between the oncology department and the nutrition unit. The existence of a nutritional consultation in oncology is highly desirable.

The nutritional monitoring of the cancer patient should be multidisciplinary and adapted to the characteristics of each center because growing evidence suggests that adequate nutritional support in these patients is of great importance. The complexity and high prevalence of different nutritional challenges make it necessary to involve specialized nutritional teams.

The primary endpoint of GC surgery is to improve survival while the role of nutritional therapy is to provide support during the perioperative period while maintaining a satisfactory level of quality of life. It seems to be difficult to provide direct evidence in the area of nutrition. However, the available evidence from scientifically satisfactory RCTs needs to be made more widely known so that nutritional therapy can be established as a multimodal treatment for GC.

Regarding future directions, we believe that the role of nutritional support during neoadjuvant/perioperative treatment of patients with GC should be the subject of intensive research. Current treatment of patients with localized GC suitable for surgery includes neoadjuvant radiation chemotherapy and perioperative chemotherapy, which are sources of catabolic stress and malnutrition. The role of nutritional support during neoadjuvant/perioperative treatment of patients with GC is expected to be more effectively defined soon.

## Figures and Tables

**Figure 1 nutrients-16-01639-f001:**
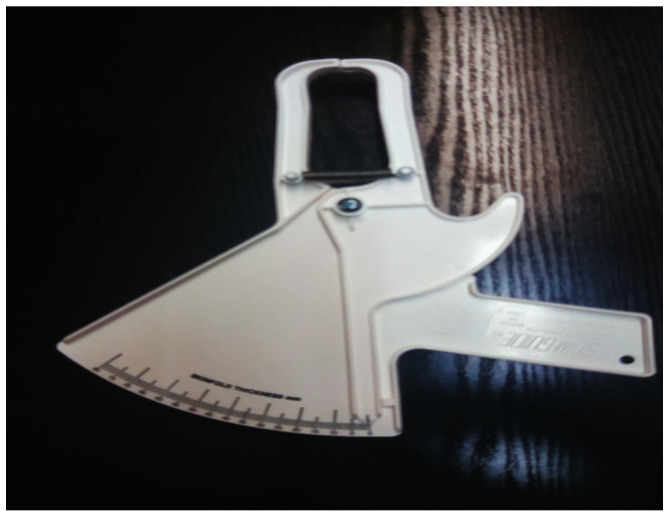
The skinfold caliper.

**Figure 2 nutrients-16-01639-f002:**
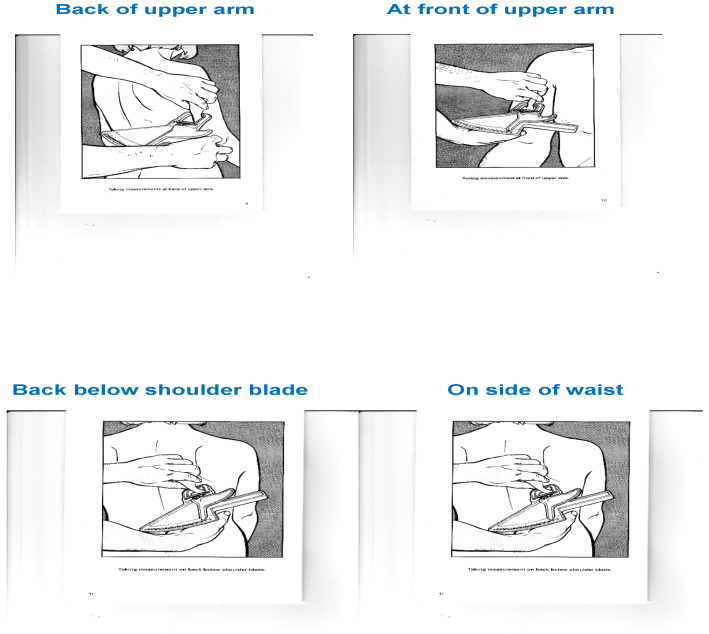
Sites of estimation of skin fold thickness.

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
