# Peer review of "Enteral Nutrition in Operated-On Gastric Cancer Patients: An Update"

_nutrients, 2024, doi:10.3390/nu16111639_

Round 1

Reviewer 1 Report

Comments and Suggestions for Authors

1. The logic of the introduction is not clear, please rephrase it.

2. In line 54-55, it is difficult to understand Therefore, there has recently been a tendency to remove as little of the stomach as possible without however this attitude to have a negative impact on the intended therapeutic effect’’.

3. In line 81-82, how to understand In this study it was shown that the determination of visceral fat area was a superior predictor compared to BMI in terms of the short-term effects of obesity, it hasn't involved with  obesity in the before text.

4. In line122-125, If the dyspeptic symptoms are adequately managed the patient is not placed on a diet poor in fat. Other dietary restrictions include avoiding high fiber intake due to its interac tion with exogenously administered pancreatic enzymes. please check it.

5. There are many inconsistent format of subheadings, Please check it.

Comments on the Quality of English Language

The manuscript needs check of English form expert

Author Response

Dear Editor,

First of all we would like to express our thanks to the reviewers for their effort in reviewing our article. Their suggestions have certainly improved its quality.

Our individual responses to the reviewers' individual comments are listed below underlined in red letters. In the revised text they also appear in red letters.

REVIEWER 1.

  1. The logic of the introduction is not clear, please rephrase it.

Answer:

The Introduction part was modified. Corrections and changes to the text are shown in the revised text in red.

  1. In line 54-55, it is difficult to understand “Therefore, there has recently been a tendency to remove as little of the stomach as possible without however this attitude to have a negative impact on the intended therapeutic effect’’.

Answer:

The text has been modified as follows to make it easier to understand (underlined in red in the revised text):

“Therefore, there has recently been a tendency to remove as little of the stomach as possible however, without this attitude to harm the intended therapeutic effect, which is the complete removal of the tumor itself’’.

  1. In line 81-82, how to understand “In this study it was shown that the determination of visceral fat area was a superior predictor compared to BMI in terms of the short-term effects of obesity”, it hasn't involved with obesity in the before text.

Answer:

The text has been modified as follows to make it easier to understand (underlined in red in the revised text):

For example Chen et al found that low BMI is associated with more severe postoperative complications and worse prognosis [6]. In the same study it was also noticed that despite a higher risk of mild postoperative complications, patients with high BMI had paradoxically better survival compared to that of patients with normal BMI [6]. In another study Yang et al found that visceral fat area was an independent risk factor for postoperative complications. They have shown that the determination of visceral fat area was represents a superior predictor compared to BMI in terms of the short-term effects of obesity [7]. It seems that unintentional weight loss is a factor of poor prognosis and mortality in cancer patients especially in those with GC

  1. In line122-125, “If the dyspeptic symptoms are adequately managed the patient is not placed on a diet poor in fat. Other dietary restrictions include avoiding high fiber intake due to its interaction with exogenously administered pancreatic enzymes.” please check it.

Answer:

The text has been modified as follows to make it easier to understand (underlined in red in the revised text):

Other dietary restrictions include avoiding a very high fiber intake because fibre may reduce pancreatic enzyme function, increase fat malabsorption, and trigger symptoms.

  1. There are many inconsistent format of subheadings, Please check it.

Answer:

The format of subheadings was modified.

Reviewer 2 Report

Comments and Suggestions for Authors

The authors have reviewed the status of enteral nutrition in gastric cancer (GC) patients.

The overall impression is that the topic of the review is of significance and that the review is well structured. However, the main problem with the manuscript is the low level of precision particularly in sections 1-3. Careful revision with addition of references is needed before a revised manuscript can be considered.    Since the manuscript is focused on the proportion of patients that undergo gastrectomy, please add this to the title. In many Western cohorts 40-50% of patients with CG do not undergo surgery due to advanced disease, while in Eastern countries 40% may be early gastric cancers that do not have similar risk of malnutrition. Please consider this and be consistent throughout the manuscript.

Some detailed questions are found below, but the authors are asked to read through an reassess the entire manuscript.

Line 34: please be more specific when stating that malnutrition rate is 60%. At which time-point? Preoperative, 30 d, 90 d postoperative? This may need some additional sentences, but it is of interest to report some numbers on the magnitude of malnutrition in an Introduction.  Despite different assessment tools the most relevant numbers would be malnutrition rates in patients undergoing gastrectomy.

Lines 55-56: Probable error “ possible without however this attitude to have …”, please adjust.

Line 62: please add references to support the statement about laparoscopic surgery is superior.

Line 63: please reconsider the phrase “the above-mentioned consequences between…”. Do the authors mean differences between laparoscopic and open gastrectomy?

Line 70: It is stated that patients with GE cancers have lost 10% of their weight before diagnosis. Please refer specifically to data on weight loss before surgery in patients who are actually undergo gastrectomy for gastric cancer.  

Section 2.1. There are several confounding factors in this field. As far as possible the authors should separate weight loss as a marker of advanced disease in patients with stage IV disease and who do not undergo surgery. They should also comment on whether studies have adjusted for old age when examining how sarcopenia is related to risk of postoperative complications. How common is significant weight loss, sarcopenia and cachexia before surgery and postoperatively?

Line 103 Please add a brief comment on the method of reconstruction and duodenal bypass; Billroth I is rarely performed in most countries.

In addition to reference 17, please add a reference to studies that are larger and do / do not demonstrate benefit.

Line 117: the rate of bacterial overgrowth is 61-75%, please provide a reference.

Lines 122-4: This sentence may be difficult to understand, please rephrase. Does reference 20 support this statement at all?

Line 130: what is “endogenous factor”? Do the authors mean intrinsic factor released by parietal cells in the gastric corpus and fundus?

Line 146: a statement about vitamin D deficiency without support of a reference. Does it concern gastrectomy overall, total or partial? Please be more specific.

Lines 149: if these statements are included in a review article, please add references. Who are the authors that recommend supplementations?

Line 154: Iron deficiency. Do the authors mean that deficiency is common after surgery? Please specify. The deficiency is probably also common at the time of diagnosis.

Section 4.2: Is this section limited to perioperative chemotherapy? Please clarify.

Comments on the Quality of English Language

No spelling mistakes detected. Some sentences unclear and need rephrasing. 

Author Response

Dear Editor,

First of all we would like to express our thanks to the reviewers for their effort in reviewing our article. Their suggestions have certainly improved its quality.

Our individual responses to the reviewers' individual comments are listed below underlined in red letters. In the revised text they also appear in red letters.

REVIEWER 2

The authors have reviewed the status of enteral nutrition in gastric cancer (GC) patients.

The overall impression is that the topic of the review is of significance and that the review is well structured. However, the main problem with the manuscript is the low level of precision particularly in sections 1-3. Careful revision with addition of references is needed before a revised manuscript can be considered.   

Answer:

We suggest that the comments of the reviewers with the subsequent corrections probably resulted in an improvement of the "precision" factor of the revised text.

Since the manuscript is focused on the proportion of patients that undergo gastrectomy, please add this to the title. In many Western cohorts 40-50% of patients with CG do not undergo surgery due to advanced disease, while in Eastern countries 40% may be early gastric cancers that do not have similar risk of malnutrition. Please consider this and be consistent throughout the manuscript.

Answer:

The title was modified according to your suggestion.

Your statement was added in the introduction part of the revised text.

Some detailed questions are found below, but the authors are asked to read through an reassess the entire manuscript.

Line 34: please be more specific when stating that malnutrition rate is 60%. At which time-point? Preoperative, 30 d, 90 d postoperative? This may need some additional sentences, but it is of interest to report some numbers on the magnitude of malnutrition in an Introduction. Despite different assessment tools the most relevant numbers would be malnutrition rates in patients undergoing gastrectomy.

Answer:

This part was greatly revised according to your suggestions. More relevant data and references were added.

Lines 55-56: Probable error “possible without however this attitude to have …”, please adjust.

Answer:

The sentence was modified.

Line 62: please add references to support the statement about laparoscopic surgery is superior.

Answer:

A relevant reference was added.

Line 63: please reconsider the phrase “the above-mentioned consequences between…”. Do the authors mean differences between laparoscopic and open gastrectomy?

Answer:

Yes. The sentence was modified.

Line 70: It is stated that patients with GE cancers have lost 10% of their weight before diagnosis. Please refer specifically to data on weight loss before surgery in patients who are actually undergoing gastrectomy for gastric cancer.      

Answer:

Relevant data were added

Section 2.1. There are several confounding factors in this field. As far as possible the authors should separate weight loss as a marker of advanced disease in patients with stage IV disease and who do not undergo surgery.

They should also comment on whether studies have adjusted for old age when examining how sarcopenia is related to risk of postoperative complications. How common is significant weight loss, sarcopenia and cachexia before surgery and postoperatively?

Answer:

The topic of weight loss was separated and analyzed as a marker of advanced disease especially in those patients with stage IV GC and patients who do not undergo surgery.

Data regarding the incidence of sarcopenia and weight loss before and after surgery were added.

Line 103 Please add a brief comment on the method of reconstruction and duodenal bypass; Billroth I is rarely performed in most countries.

Answer:

A brief comment regarding the methods of reconstruction was added.

In addition to reference 17, please add a reference to studies that are larger and do / do not demonstrate benefit.

Answer:

A relevant statement was added

Line 117: the rate of bacterial overgrowth is 61-75%, please provide a reference.

Answer:

The relevant reference was provided.

Lines 122-4: This sentence may be difficult to understand, please rephrase. Does reference 20 support this statement at all?

Answer:

The sentence was rephrased. Reference 20 deleted. 

Line 130: what is “endogenous factor”? Do the authors mean intrinsic factor released by parietal cells in the gastric corpus and fundus?

Answer:

Certainly! The term was changed to intrinsic factor.

Line 146: a statement about vitamin D deficiency without support of a reference. Does it concern gastrectomy overall, total or partial? Please be more specific.

Answer:

A relevant reference [24] was added. The statement is referring to operated patients.

Lines 149: if these statements are included in a review article, please add references. Who are the authors that recommend supplementations?

Answer:

These statements are included in the following systematic review: Muszyński T, Polak K, Frątczak A, Miziołek B, Bergler-Czop B, Szczepanik A. Vitamin D-The Nutritional Status of Post-Gastrectomy Gastric Cancer Patients-Systematic Review. Nutrients. 2022;14:2712. doi: 10.3390/nu14132712. PMID: 35807892; PMCID: PMC9268678. This reference was added in the list of references.

Line 154: Iron deficiency. Do the authors mean that deficiency is common after surgery? Please specify. The deficiency is probably also common at the time of diagnosis.

Answer:

Iron deficiency is common after surgery. However iron deficiency and anemia are common at the time of diagnosis being among the most important alarming laboratory findings.

Section 4.2: Is this section limited to perioperative chemotherapy? Please clarify.

Answer:

This section applies to all patients undergoing chemotherapy regardless of the stage of the disease and the surgical or non-surgical treatment. This has been clarified in the revised text.

Round 2

Reviewer 2 Report

Comments and Suggestions for Authors

The authors have revised the manuscript well.

Comments on the Quality of English Language

Some capital letters within sentences are unconventional. Please consider adjusting while proofreading.